# Investigations on Backflush Cleaning of Spent Grain-Contaminated Filter Cloths Using Continuous and Pulsed Jets

**DOI:** 10.3390/foods11121757

**Published:** 2022-06-15

**Authors:** Roman Alejandro Werner, Alexander Michael Hummel, Dominik Ulrich Geier, Thomas Becker

**Affiliations:** Chair of Brewing and Beverage Technology, School of Life Sciences, Technical University of Munich, Weihenstephaner Steig 20, 85354 Freising, Germany; alex.hummel@gmx.net (A.M.H.); dominik.geier@tum.de (D.U.G.); tb@tum.de (T.B.)

**Keywords:** cleaning, filter media, filter cloth, backflush cleaning, pulsed wash jets, spent grains

## Abstract

This study investigated the continuous and pulsed backflush cleaning of woven fabrics that act as filter media in the food and beverage industry. Especially in breweries, they are commonly used in mash filters to separate solid spent grains from liquid wort. After filtration, the removal of such cereal residues via self-discharge is necessary. However, this filter cake discharge is typically incomplete, and various spots remain contaminated. In addition to the reduced filter performance of subsequent batches, cross-contamination risk increases significantly. A reproducible contamination method focusing on the use case of a mash filter was developed for this study. Additionally, a residue analysis based on microscopical image processing helped to assess cleaning efficiency. The experimental part compared two backflushing procedures for mash filters and demonstrated fluid dynamical, procedural, and economic differences in cleaning. Specifically, pulsed jets show higher efficiency in reaching cleanliness faster, with fewer cleaning agents and less time. According to the experimental results, the fluid flow conditions depended highly on cloth geometry and mesh sizes. Larger mesh sizes significantly favored the cloth’s cleanability as a larger backflush volume can reach contamination. With these results, cloth cleaning can be improved, enabling the realization of demand-oriented cleaning concepts.

## 1. Introduction

Filter cloths are textiles widely used in many solid–liquid filtration processes. Mono-or multifil yarns as chaining and shoot threads form a particular fabric’s weave pattern and create different surface topographies [1]. The term filter geometry can be derived from a specific filter structure, significantly influenced by mesh sizes and weave type. During filtration, different apertures represent the selectivity of particle sizes [2]. In addition, the surface properties of materials should be considered, as these can influence the selective binding of certain substance classes on filter cloth [3]. The materials used are mostly polymers such as polypropylene or polyamide. Threads of various thicknesses are produced by technical extruders from polymer raw materials. In addition to different mesh sizes, interwoven threads have hilly structures that form flow channels of different depths and facilitate particle adhesion points. This intricate geometry facilitates filtration processes and provides various application possibilities [4]. However, the complexity and material limits (e.g., temperature and pH value) can be detrimental to the filter surface during the cleaning process [5]. The structures of commercially available filters contradict the common hygienic design principles. To date, no corresponding guidelines for filter cloths have been published by corresponding organizations or associations [6].

Filter cloths have a wide range of applications in the beverage, food, and pharmaceutical industries. In breweries, mash filters, which are chamber or membrane filter presses, use filter cloths to separate solid spent grains from the liquid wort. This segregation can also be performed using lauter tuns with a slotted plate as filter support and spent grain as the actual filter medium. The plate is usually located close to the tun’s bottom and facilitates dead-end filtration. Unfortunately, the lauter tun’s flexibility is limited due to long filter times as the entire mash volume has to be separated using one single tun. Here, mash filter systems are often preferred because they are typically more flexible in batch quantities and cereal variety (e.g., unmalted barley or corn) than other systems [7]. Such filter presses consist of several frames covered with large-scale filter cloths, forming filter plates. The compression of various plates leads to multiple filter chambers in the system [8]. After filtration, the release and pulling apart of these plates produce a cake discharge of spent grains from the filter cloth. However, this cake release is typically incomplete, as spent grain-contaminated spots remain on the cloth and in its meshes.

After a certain batch amount, extensive cleaning must be performed to ensure sufficient spent grain removal and high product safety. In particular, the residues in the meshes create a problem during cleaning. A lack of optimization is evident in the brewing industry, as cleaning is commonly performed manually or only with rigid cleaning systems. Due to increased ecological and economic awareness over the last decades, the beverage and food industries (especially breweries) have also highlighted the necessity to enhance cleaning concepts. Modern cleaning research has recognized the high potential in more optimized mechanical cleaning concepts and simultaneous chemical agent reduction [5,9,10,11,12].

One promising cleaning method for mash filter systems can be backflush cleaning, which has already enhanced the cleaning of membranes in dairy or wastewater treatment. Here, this technique pushes water from the filtrate side through the filter medium. Concerning cloths, the aim is to realize sufficient mechanical forces in adjusting flow rates to release adhesive forces between spent grains and the filter cloth at the feed side. In addition, the backwashing must ensure that the residues in the mesh can be reached. This may be an advantage compared with forward flushing as backflushes can reach small particles that blind the filter meshes more efficiently and enable direct dirt transport away from the filter cloth. However, many aspects are still unknown and require comprehensive research. Objects for investigation are the flow loss due to pressure drops and a possible cleaning effect decrease during the cloth passthrough. Here, a loss of mechanical effects can occur that needs to be compensated.

Weidemann et al. [13] and Leipert and Nirschl [14] conducted the first studies on pulsed backflushing filter cloths using model particles. Other publications generally proved the suitability of discontinuous streams in cleaning concepts results [15,16]. Focusing on pulsed cleaning in an industrial context, previous publications [10,13,17,18,19,20,21,22,23,24] have shown promising results using this cleaning method in beverage and food applications such as pipes or heat exchangers. In dust filters and gas filtration, pulsed cleaning is already state of the art [25,26,27,28]. In these studies, the pulsed mode significantly increased the mechanical component of the cleaning concept.

The first studies on cleaning spent grains loaded filter cloth with forward flushes were conducted by Morsch et al. [11,12,29] and Werner et al. [30]. Their conclusions were a promising starting point for investigating the industry applications of pulsed backflushes and breaking down decisive mechanisms in filter cloth cleaning. An in-depth scientific analysis was necessary as different disciplines need to be combined, such as biophysics (heterogenous contamination), fluid dynamics (backflush), material science (filter cloth), and process engineering (cleaning and process design). In the end, in brewery practice, the technical realization of this cleaning concept can be achieved either by nozzles arranged on the back of the plate or by direct backwashing of the filter cloth with pumps.

This study aimed to use pulsed jets to enhance the mechanical cleaning effects of backflushing filter cloth. Two different filter cloths with different mesh sizes and weave types formed the centerpiece of the investigations. An industrial-close use case was selected, focusing on spent grains-loaded filter cloths that appear in mash filters in a brewery. For this purpose, a tailor-made contamination method was developed to assess cleaning procedures reproducibly. This was combined with a novel residue analysis by microscopical image processing. The experimental results illustrated the most influential cloth cleaning parameters on backflushing and determined the most suitable cleaning design between continuous and pulsed backflushing. The latter used pulsed streams with a defined ratio of jet lengths to pauses, increasing mechanical cleaning effects. In addition, using water as the only cleaning agent significantly reduced the amount of aggressive chemicals possible. The increase in mechanical parameters may also decrease other factors such as temperature and time. This change in the cleaning method can preserve cloth materials, extend the filter lifetime, and reduce costs.

## 2. Materials and Methods

### 2.1. Filter Cloths

The experimental investigations used two different types of filter cloths. Both were manufactured by Sefar AG (Heiden, Switzerland), and their names are 05-1001-K 120 (here: F1-TWL-80/100) and 03-1010-SK 038 (here: F2-STN-20). The annotation F1 represented filter type 1 with a twill weave (TWL) and mesh sizes of 80 and 100 µm. On the other hand, filter type 2 was designated as F2 and had a satin weave (STN) and mesh sizes of 20 µm. They are both commonly used in different filtration applications and mash filters, in particular. Table 1 illustrates the different filter properties and design features. The main differences between the two filter cloths were the material, the weave type, and the mesh size.

A detailed consideration of the fluid dynamics of backflushing led to the assumption that the mesh sizes will significantly influence cleaning efficiency. Here, F1-TWL-80/100 had mesh four times larger than that of F2-STN-20, whereas the thicknesses of both cloths were similar. Additionally, the filter cloths had calendared surfaces, as the fabric surface had been smoothed by rolling through heated rollers. Figure 1 shows both filter types, with images taken using the digital microscope Keyence Corporation VHX 2000D (Osaka, Japan). For this purpose, the filter cloth was positioned directly under the microscope lens for frontal and side imaging. The objective was used with a magnification factor of 50 to display a sufficient degree of detail. For the subsequent contamination process, the filters were cut into rectangular samples with a length of 4.4 cm and a width of 1.4 cm. This size enabled a suitable contamination area that was able to be cleaned in the cleaning device and examined by the digital microscope VHX 2000D.

### 2.2. Contamination Preparation

In this study, the mash filter in a brewery served as an industry-related case study that required a corresponding contamination type based on spent grains. Thus, a standardized mash using the brewer’s MEBAK R-206.00.002 [31] congress mashing method provided a suitable contamination suspension. However, a slight adaption of the method was necessary. In the developed procedure, the cross-hammer mill PerkinElmer, Inc. LM 3100 (Hamburg, Germany) was used for crushing the malt instead of a roller mill. In general, hammer mills are the preferred type of mill in breweries to grind malt for mash filter applications. The malt used was a Pilsener Malt of the malthouse Malzfabrik Mich. Weyermann GmbH & Co. KG^®^ (Bamberg, Germany), the most commonly used malt type in Germany. The resulting spent grains of this malt contained a wide variety of non-cellulosic polysaccharides, proteins, cellulose, lignin, lipids, and ash that produced problematic sticky or insoluble contamination on filter cloths [32,33]. The particle size distribution q3 ranged from >5 μm to <1000 μm, where the most significant proportion was from 20 μm to 225 μm. For this reason, applying simple and homogenous model contamination (e.g., particles) to assess industrial cleaning processes has to be regarded critically.

Moreover, the husks of the barley malt were interesting for contaminating filter cloths. Husks are insoluble parts of the barley shell that a hammer mill crushes into small particles. Once these husk parts get stuck in filter meshes, they are difficult to remove by standard cleaning methods (e.g., chemicals or surface rinsing).

The congress mash procedure started with mixing 50 g of malt grist and 200 mL of demineralized water, preheated to 45 °C. The mixing was carried out with care and under constant stirring at 90 rpm to avoid clumping. This stirring speed was kept constant by all following steps. Subsequently, the mash was held at a temperature of 45 °C for 30 min and then heated up to 70 °C. After reaching this temperature level, another 100 mL of demineralized water (70 °C) was added and mashed for another 60 min. In a final step, the mash was cooled to 20 °C and weighed up to 450 g with demineralized water (see Figure 2A).

### 2.3. Contamination Application

Before the contamination suspension could finally be applied, an aluminum ring with an inner diameter of 10 mm was placed on each filter cloth sample (see Figure 2B). This stencil specified the dimensions of the circular contamination and thus enabled a standardized spent grains spot for the cleaning experiments. After prewetting the cloth samples, a mash volume of 200 µL was pipetted in this stencil to generate the contamination spot. Subsequently, the filter cloths were left to rest for 4 h at 20 °C, corresponding to an average filtration time with subsequent processing and set-up time in industrial practice. After this defined adhesion time, the aluminum circle was removed, leading to a reproducible standardized contamination spot with the same cake surface (diameter 10.0 mm) and thickness (approx. 2.6 mm).

Thus, the reproducible contamination spots provided an ideal starting point for the cleaning experiments. Due to the consistently equal mash volume, the contaminated cloth areas coincided in size and height and showed similar cleaning behavior. The reproducibility was mainly achieved by the standardized contamination procedure, same-time intervals, and equal particle size distribution. In particular, the applied hammer mill proved to be a significant advantage. In this grinding apparatus, the malt grains were milled into small particle sizes, which in this case contributed to a more homogeneous contamination appearance.

### 2.4. Cleaning Device and Cleaning Processes

The contaminated filter cloth sample was inserted into a holed microscope slide, and an image of the contaminated status quo was acquired (see Figure 2C–E). This procedure was necessary to assess the cleaning degree in a later step. Subsequently, the cloth sample in the object slide was placed directly in the cleaning apparatus (see Figure 2F). The sample was inserted with the filtrate side towards the cleaning nozzle, which ultimately allowed the cloth to be backwashed.

The experimental cleaning setup had a defined stream channel positioned vertically to simulate real-world conditions in an industrial mash filter system (see Figure 2G). This arrangement enabled industry-like conditions to be reflected in a filter press. Above the filter cloth, a cell lid with an inserted cylindrical nozzle was mounted. The nozzle streamed vertically (incidence angle = 90°) on the filtrate side of the filter cloth sample on the microscope slide, backflushing the filter cloth. The nozzle had a pipe connected to a solenoid valve, where cleaning streams with defined time lengths could be adjusted by an Arduino© NANO microcontroller of Arduino LLC (Boston, MA, USA). The adjustable parameters on the cleaning device can be taken from Table 2. The experimental cleaning setup enabled the generation of pulsed jets with variable pulse lengths and pauses as well as standard continuous cleaning. As the mechanical cleaning effect of backflush cleaning was the focus, demineralized water was used as the only cleaning agent in all experiments.

The wash water was directly transportable from a storage tank to the cleaning device by applying pressure to the tank. Hence, the velocity of the fluid was directly affected by the pressure level. The connected solenoid valve was connected to an upstream 3-way system. By closing the valve, the water circulated in and out of the tank. When the valve was open, water shot into the nozzle. This jet pre-acceleration was necessary to achieve a constant and adjustable speed. Before every measurement, it was necessary to determine the jet velocity. Key parameters included the mass of the emitted water *m_Fluid_* and the corresponding time length *t_open_*. Equation 1 theoretically calculated the desired impact velocity by weighing the emitted fluid mass.
(1)$$v_{Nozzle}=\frac{m_{Fluid}}{t_{open}\times A_{Nozzle}\times \rho_{Fluid}}$$

Here, the amount of water *m_Fluid_* (measured gravimetrically) escaping from the nozzle at the surface *A_Nozzle_* (1.57 × 10^−3^ m^3^) equaled the velocity *v_Nozzle_*. For an exact calculation, the water density *ρ_Fluid_* (1.0 g·cm^−3^ at 20 °C) and a defined nozzle-opening time of 10 s were required. The cleaning temperature was 20 °C and kept constant in the test laboratory.

## 3. Results and Discussion

### 3.1. Residue Analysis

A suitable detection method for the residues was the critical element in assessing the cleaning degree of the corresponding cleaning experiment. For this purpose, the digital microscope VHX 2000D was used to capture images of the filter cloths. The microscope camera (18 MP) took images of the contaminated and cleaned filter cloths with a 20× magnification. An image analysis algorithm coded in MATLAB 2019b of The MathWorks^®^, Inc. (Natick, MA, USA) processed and analyzed both pictures (see Figure 3). For the image processing, the color space HSV was used.

The algorithm generated a cut picture and converted it into hue (H), saturation (S), and value color space (V) (see Figure 3b). The determination of the contaminated areas depended on the desired color spectrum in H and the exclusion of white and gray tones by setting the saturation scale. For this type of residue analysis, there had to be sufficient color contrast between the filter cloth and contamination spots. The light conditions were adapted to the measurement environment in the lab. The proper adjustment implied the balanced illumination of the image while avoiding reflections on the contamination in case of overlighting.

Furthermore, the lighting settings had to be consistent to achieve highly reproducible measurements. In the case of spent grain-loaded filter cloths, the color values of contamination spots were brownish, ocher, and yellowish, allowing sufficient detectability on white filter cloths. The precisely adjustable illumination of the samples was achieved with the aid of a high-power LED lamp installed in the Keyence microscope.

Subsequently, the resulting binary image’s contaminated areas (white) were computed. By identifying the contaminated areas before cleaning *A*_1_ and after cleaning *A*_2_, the cleaning degree *G* was determined with Equation (2)
(2)$$G\;\left[-\right]=1-\frac{A_{2}}{A_{1}}$$

The cleaning degree reflected the amount of contamination removed from the respective cleaning process. Thus, the efficiency of cleaning methods depended on the cleaning degree.

### 3.2. Determination of the Backflow Factor

Before conducting the parameter variation and comparing the different cleaning concepts, it was necessary to consider the resulting cleaning effect in detail. For this reason, it was necessary to identify the relevant backflush volumes that can percolate the filter cloth. Ultimately, this specific volume generated the cleaning effect.

Here, the novel introduced backflow factor *ζ_BF_* could represented the backflush’s volume (the portion of the total amount of water used) that could flow through the dense filter meshes. The other portion was deflected on the cloth and flowed off laterally on the filtrate side due to back pressures at thread bridges. *ζ_BF_* was the guiding parameter for the amount of water passing through the filter. For this purpose, the filter sample was mounted on a microscope slide and placed in front of the nozzle. A seal ensured the complete separation between the top and bottom parts of the slide. Afterward, a jet with mass *m_Fluid_* was streamed onto the filter cloth. The amount of water *m_Fluid_Filtered_* that passed through the filter cloth was drained into a small container. The amount of fluid passing through the filter was measured gravimetrically and helped determine *ζ_BF_* with Equation (3).
(3)$$\zeta_{BF}=\frac{m_{Fluid\_Filtered}}{m_{Fluid}}$$

Figure 4 shows the results of *ζ_BF_* for both filter cloth types using six different stream velocities with an inflow time of 10 s.

Regarding F1-TWL-80/100, the results depicted a slightly decreasing *ζ_BF_* by increasing the inflow velocity *v_Nozzle_*. This was in contrast to the observation regarding F2-STN-20, where a slight increase was visible. On the one hand, *v_Nozzle_* significantly influenced wash-water consumption, increasing with higher velocities. On the other hand, the difference was insignificant, which derived the assumption that *v_Nozzle_* had no increasing influence on *ζ_BF_* when only short cleaning streams were used. This observation may result from the free passage surface, which allowed only a certain amount of backflush to pass in a particular time interval. The passage surface depended on the mesh sizes, weave type, and thread thickness.

However, there were significant differences in a direct comparison of both filter types. For F1-TWL-80/100, twice as much wash water shot through the filter cloth. This result depended on mesh sizes, which in F1-TWL-80/100 were four times bigger. For F2-STN-20, there was a larger filter surface, resulting in back pressures and increased dynamic pressures. Overall, the measured values of F2-STN-20 were only approximately half those of F1-TWL-80/100.

### 3.3. Comparison of Pulsed and Continuous Backflush Cleaning

Jet cleaning can be applied either as a standard continuous or the novel pulsed cleaning process. Recently, the pulsed procedure has been increasingly considered for filter cloths [10,13,14,17,30]. Figure 5 compares the continuous and pulsed backflushing techniques using the velocities 2 m/s and 4 m/s for cleaning both filter cloths.

Significant differences were observable in the results regarding a low velocity of 2 m/s. Pulsed jets resulted in significantly higher cleaning degrees when compared to the two cleaning methods. Notably, the transition zone from 32 pulses/32 s to 256 pulses/265 s showed these efficiency differences. In summary, the pulsation operating mode had a high ecological and economic potential, as comparable degrees of cleaning could be achieved in a shorter time with considerably less cleaning fluid than the continuous operation. Regarding the applied cleaning velocities, a difference between 2 and 4 m/s became apparent, which is highlighted more in Section 3.4.

The comparison of both filter cloths also showed efficiency differences. The better cleanability of F1-TWL-80/100 confirmed the above-stated assumptions. Fewer pulses or small cleaning lengths did not lead to appropriate cleaning degrees, resulting in poor removal effects. The appropriate adjustment was in the mentioned transition zone and had significantly higher pulse numbers and washing lengths depending on the filter geometry. Pulsed cleaning had more advantages when selecting the most suitable process design. Augustin et al. [18] also saw a higher efficiency of pulsed cleaning, resulting in higher wall shear stress and waviness on the applied technical surfaces, favoring contamination detachment.

Furthermore, in this experimental setup, the filter cloth was positioned vertically. Respective pulse pauses allowed improved drainage of the wash water and simultaneous contamination removal from the cleaning zone. A reduction of the liquid layer thickness also occurred, which could have cushioned contamination from previous jets.

### 3.4. Influence of Cleaning Velocity

The previous results indicated an influence of the stream velocity *v_Nozzle_* on the cleaning degree. Other studies [21,34,35,36] also followed that the dirt-removing wall shear stress depended on the applied flow velocity. A closer look at the transition zone needed to be taken regarding the velocity influence: for F1-TWL-80/100, the transition area was 32 pulses/32 s, whereas it was 128 pulses/128 s for F2-STN-20. As adjusted velocities, a range from 1.5 to 4.0 m/s was chosen as these are standard cleaning settings in the beverage and food industry [37,38]. Figure 6 shows the influence of these average velocities on the cleaning degree of both filter cloths.

The results depicted in all subfigures showed increased cleaning degrees using higher velocities *v_Nozzle_*. The comparison of both cleaning designs showed pulsed jets reaching similar cleaning degrees earlier. From the velocity curves in each diagram, the cleaning degree remained constant at a small value until a particular velocity was applied. Afterward, a linear increase in the cleaning degree could be assumed.

Of high interest was the velocity at which the degree of cleaning increased significantly. At this point, the cleaning efficiency could be related to the turbulence degree. Efficient cleaning procedures are facilitated by turbulent flows, increasing the wall shear stress, reducing the boundary layer, and absorbing more contamination [37]. In particular, the results of F1-TWL-80/100 showed a significant cleaning effect with adjusting inflow velocities of 2.5 to 3.0 m/s. It could be assumed that a turbulent backflush at the contaminated filter side only occurred with these higher velocities at the inflow side.

Thus, adjusting a higher *v_Nozzle_* is necessary to reach enough mechanical effect and turbulences of the backflush, favoring the cleaning process. Regarding continuous cleaning (32 s) of F1-TWL-80/100, this issue became apparent at 3.5 m/s. Concerning the pulsed method, the transition point is visible at 3.0 m/s. In addition, F2-STN-20 showed similar observations. Here, a four-times longer cleaning length and pulse number were necessary to reach the corresponding transition point. Comparing continuous and pulsed backflushing, the point of interest was 2.0 m/s. Here, the pulsed jet achieved 1.5 times higher cleaning degrees.

In conclusion, the adjusted velocity *v_Nozzle_* of backflushing was crucial to reaching acceptable cleaning efficiency. Remarkably, the same cleaning effects on the filtrate side could not necessarily be assumed when setting a turbulent cleaning jet on the feed side. However, pulsed cleaning jets could help achieve a better cleaning result, even with lower backflush velocities. Firstly, pulsed cleaning can compensate arising losses in the cleaning effect and reduction of turbulent conditions. Secondly, there was a significant ecological and economic potential in using pulsed backflushes, as cleaning time and fluid consumption in particular could be reduced.

### 3.5. Comparison of Both Filter Types

In the last step, both filter cloths were compared to illustrate the differences in their cleanability. For this purpose, two different velocities were used to examine the influence of filter geometry. The first velocity was the lowest at 1.5 m/s, where all mechanical effects were minimal, helping to understand the different cleanabilities. The second cleaning speed was at a change to the turbulent condition of 3.5 m/s. All results are shown in Figure 7.

A difference between both filters was visible in all diagrams. The abrupt increase in the degree of cleaning above a certain speed was again observable. In addition, the transition zone became obvious between 16 pulses/16 s to 128 pulses/128 s. After the contamination application, spent grain cake discharges penetrated the meshes due to the filtration effect and dried partly. The jets rewetted the contamination in these first cleaning steps, reducing adhesion towards the cloth surface.

Regarding the geometry, F1-TWL-80/100 was a twill weave, whereas F2-STN-20 was a satin weave. According to Table 1, F1-TWL-80/100 had a simpler geometry and bigger mesh sizes that facilitated backflush cleaning. The difference in mesh sizes had a decisive influence on the cleaning performance. Larger mesh sizes allowed much larger quantities of washing water to pass through. This larger volume of wash water was thus able to remove more significant amounts of contamination and ultimately transport it away from the contamination zone. In addition, an influence on the backflush’s velocity could also be observed here, which was reduced to a minor degree in the case of large-mesh cloths.

## 4. Conclusions

This study presented filter cloth backflushing for industrial-close contamination removal—in this case, spent grains in a brewery. Therefore, continuous and pulsed jets were applied and compared. The observations help to clarify backflush cleaning and oversee critical aspects of finding the proper cleaning procedure. The method showed satisfactory reproducibility for industry-related food contamination based on spent grains, which was confirmed by the determined confidence intervals. The results also show significant differences when comparing the different cleaning concepts and critical cleaning parameters.

Firstly, pulsed backflushing could be presented as a promising cleaning concept for filter cloths and mash filters in particular. The increased mechanical cleaning effect could enhance and optimize the cleaning process significantly. Pulsed backflushes were advantageous in economic and ecological aspects as they cleaned more effectively, achieving similar cleaning degrees faster, with a reduced amount of wash water and less cleaning time. However, at this point, it needs to be clarified that chemical agents will always be necessary to guarantee hygienic and microbiological-uncritical conditions.

Secondly, the mesh size and respective backflow factor *ζ_BF_* influenced cleanability, as larger meshes allowed more cleaning fluid to pass the filter cloth. The weave type concerning mesh sizes was decisive in a filter cloth’s cleanability in this context. Other parameters such as materials or calendaring seemed to be less important. In addition, the latter has to be critically discussed as it can, even more, reduce mesh sizes by cloth flattening. Conclusively, filter geometry—derived from weave type and mesh sizes—is the decisive parameter in designing the appropriate backflushing concept. More fine meshes and complex geometries require higher inflow velocities to achieve a sufficient cleaning effect.

Ultimately, the cleaning velocity was the third vital aspect in finding the proper cleaning concepts. Higher inflow velocities resulted in higher cleaning degrees, although their speed seemed to be reduced after passing through the filter cloth. Thus, adjusting a sufficient nozzle velocity *v_Nozzle_* was crucial to ensure sufficient cleaning effects after the passage. Before selecting the respective cleaning parameters, a cloth-specific transition zone from low to high cleaning degrees had to be identified. However, higher velocities also lead to higher impulses, damaging the cloth and demanding more cleaning liquid.

In summary, it can be asserted that pulsed cleaning jets are preferable for mash filter cleaning. Depending on the cloth design, applying 64 pulses in a quick sequence (clocking 100 ms) and an incident flow velocity of at least 3.0 m/s can yield good cleaning results. In addition, a more apertured cloth favored cleaning with backflushes. However, this depends on the application field and the required filtration properties. Since relatively large particles are separated during mash filtration, cloths with larger mesh sizes can be used well here.

For the food industry—especially breweries—pulsed backflush cleaning is a promising technology to optimize filter cloth cleaning in mash filters. In addition, the knowledge gained can optimize filter press cleaning in particular. The results of this study can also pave the way from rigid, operator-based cleaning procedures to demand-oriented, digitalized concepts that enable modern cleaning in terms of Industry 4.0. For a final transfer of the results into industrial application, the backflushing has to be technically realized on a large-scale. Here, direct backflushing of the entire filter cloth surface with a pump system is conceivable. It is also possible to place nozzles in the filter plate behind the filter cloth. If appropriately arranged, they enable the targeted generation of backflushes with an efficient cleaning effect. The choice of the proper technique always depends on the design of the filter press, the manufacturer, and the prevailing local requirements. For an efficient cleaning process, the cloth geometry applied should, in all cases, be taken into consideration in the preliminary stages. This requirement largely which basic cleaning parameters must be used.

## Figures and Tables

**Figure 1 foods-11-01757-f001:**
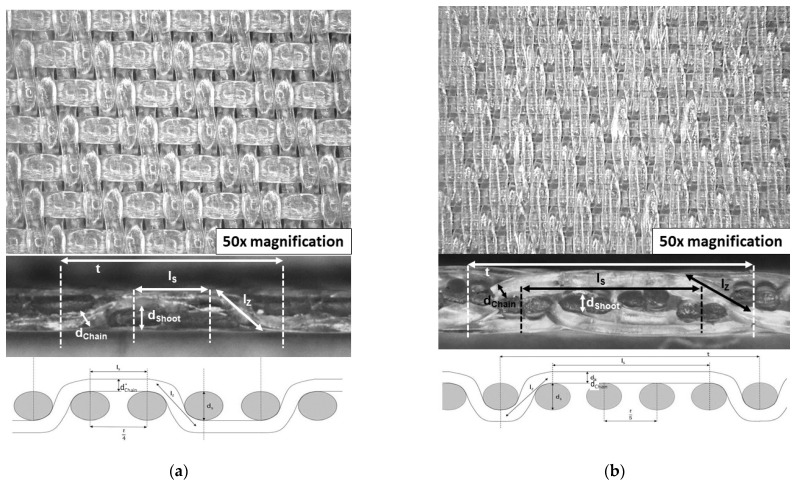
Microscopic image as top and side view and schematic drawing of the filter cloths: (**a**) F1-TWL-80/100; (**b**) F2-STN-20; *t* = length to the repetition on the chain/shoot thread; d_Chain/Shoot_ = diameter of chain/shoot thread; l_S_ = length straight-thread section; l_Z_ = length slanted-thread section.

**Figure 2 foods-11-01757-f002:**
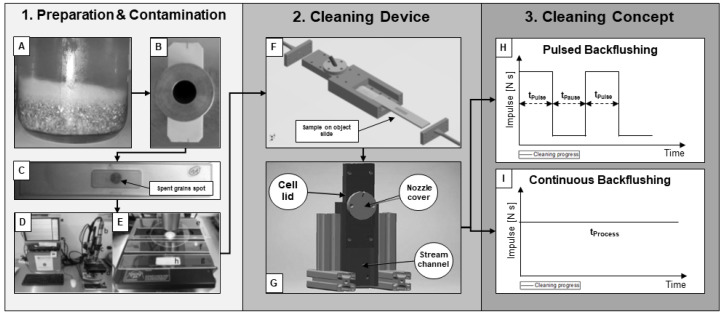
Illustration of the complete experimental procedure. (**A**) Prepared congress mash. (**B**) Aluminum stencil on cloth sample. (**C**) Cloth sample in the object slide. (**D**) Digital microscope device. (**E**) Object slide under microscope lens, identifying contamination. (**F**) Insertion of object slide into cleaning device. (**G**) Final vertical positioning of the cleaning device. (**H**,**I**) Applied cleaning concept. This illustration concludes Section 2.2, Section 2.3 and Section 2.4.

**Figure 3 foods-11-01757-f003:**
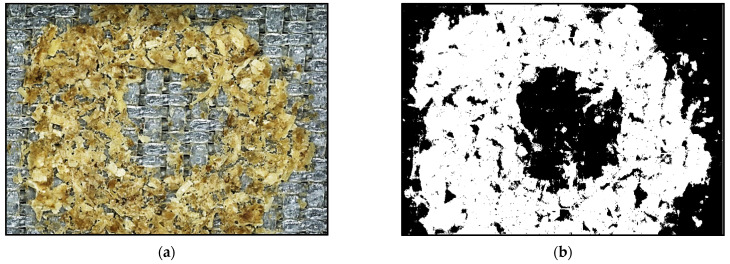
Original image of the cleaned filter sample (**a**) and its binarized image (**b**). The white spots represent the remaining contamination after cleaning, whereas the black areas show the cleaned or contaminated surface. The picture also illustrates the area where the jet streamed through the filter cloth (the black spot in the middle of (**b**)).

**Figure 4 foods-11-01757-f004:**
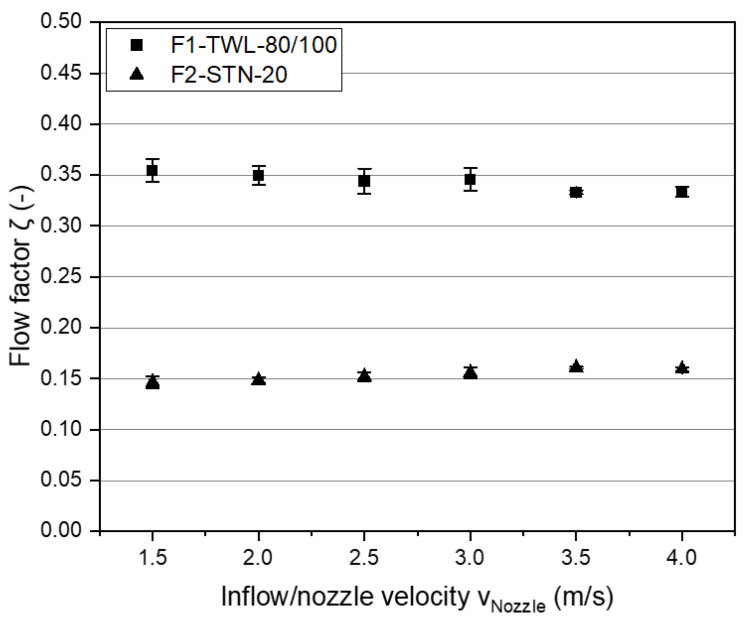
Backflow factor *ζ_BF_* of a defined water volume streaming through filter cloths. Confidence intervals with α = 0.05.

**Figure 5 foods-11-01757-f005:**
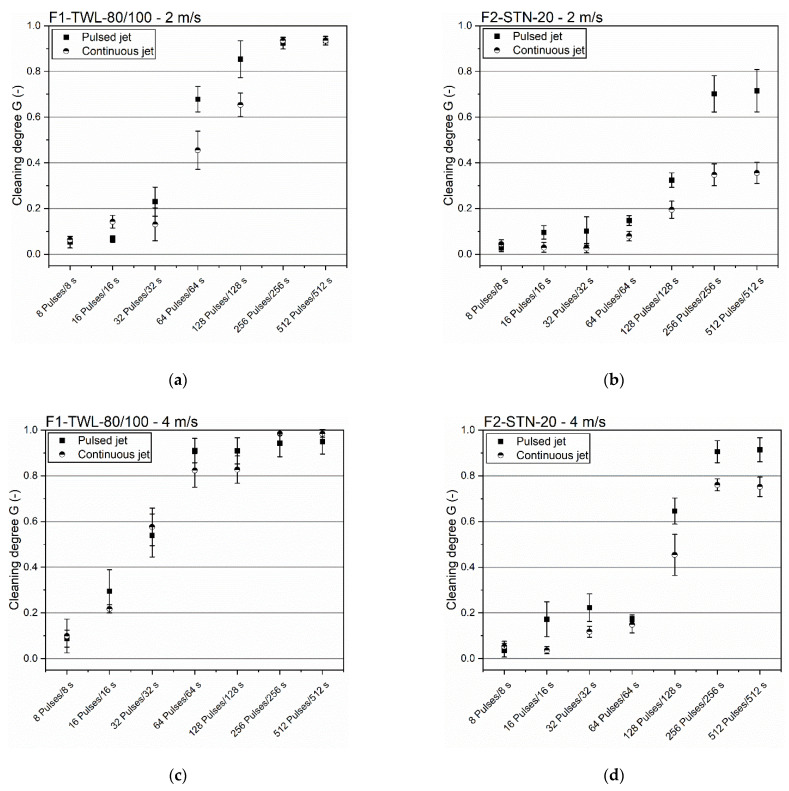
Comparison of pulsed and continuous cleaning with two different cleaning velocities. Settings for pulsed cleaning: t_Pulse_ = t_Pause_ = 0.1 s in a logarithmic interval [8:265]. Continuous cleaning settings: unpaused in a logarithmic interval [8:256] s. (**a**) F1-TWL-80/100 and 2 m/s. (**b**) F2-STN-20 and 2 m/s. (**c**) F1-TWL-80/100 and 4 m/s. (**d**) F2-STN-20 and 4 m/s. *n* ≥ 5. Confidence intervals with α = 0.05.

**Figure 6 foods-11-01757-f006:**
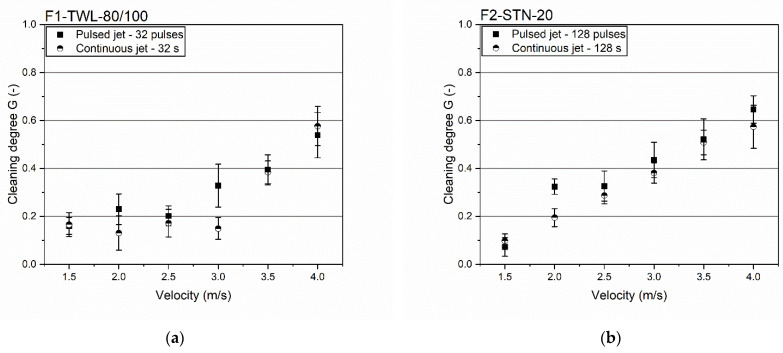
Velocity variation of *v_Nozzle_* regarding the transition area of pulsed and continuous cleaning. (**a**) Pulsed wash jet with 32 pulses and continuous wash jet with a length of 32 s at F1-TWL-80/100. (**b**) Pulsed wash jets with a quantity of 128 pulses and continuous wash jet with a length of 128 s at F2-STN-20. t_Pulse_ = t_Pause_ = 0.1 s. *n* ≥ 5. Confidence intervals with α = 0.05.

**Figure 7 foods-11-01757-f007:**
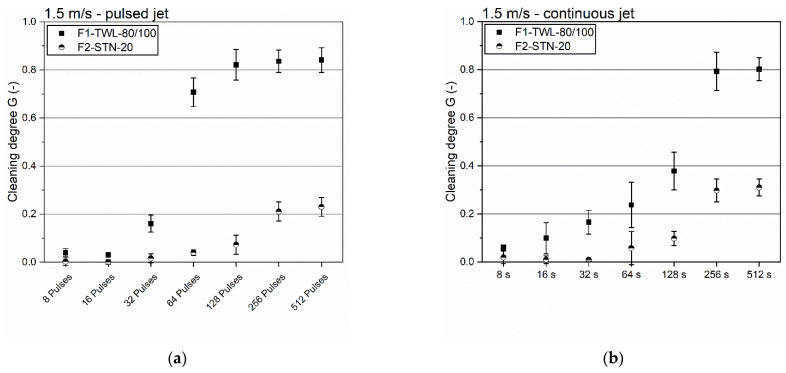
Comparison of both filter cloths using pulsed and continuous cleaning. (**a**) Pulsed cleaning at an inflow velocity of 1.5 m/s. (**b**) Continuous cleaning at an inflow velocity of 1.5 m/s. (**c**) Pulsed cleaning at an inflow velocity of 3.5 m/s. (**d**) Continuous cleaning at an inflow velocity of 3.5 m/s. *n* ≥ 5. Confidence intervals with α = 0.05.

**Table 1 foods-11-01757-t001:** Applied filter cloth types with their distinct properties and designations.

Designation	Manufacturer’s Designation	Mesh Sizes [µm]	Material	Weave Type	Thickness [µm]	Finish
F1-TWL-80/100	05-1001-K 120	80 & 100	Polypropylene	TWL	480	Calendared
F2-STN-20	03-1010-SK 038	20	Polyamide 6.6	STN	470	Special calendared

K = calendared. SK = special calendared. TWL = twill weave. STN = satin weave.

**Table 2 foods-11-01757-t002:** Cleaning parameters and the corresponding ranges applicable in the cleaning setup.

Cleaning Parameter	Range	Unit
Jet velocity	1.5–4.0	m/s
Number of pulses	1–1000	-
Minimum pulse length	15	ms
Standard pulse–pause ratio	100/100	ms/ms

## Data Availability

The data are available on request from the corresponding author.

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
