# Peer review of "Investigations on Backflush Cleaning of Spent Grain-Contaminated Filter Cloths Using Continuous and Pulsed Jets"

_foods, 2022, doi:10.3390/foods11121757_

Round 1
Reviewer 1 Report
The manuscript has great relevance for the academy, mainly due to the novelty of the theme related to filter cloths. The increasing economic awareness of the food and beverage industries, as emphasized by the authors, has led to new demands related to the need to improve cleaning concepts. The results of this manuscript contribute to these current demands.
1. I suggest that authors review the bibliographic references used, since more than half of the references were published more than ten years ago.
2. I would like the authors to detail the conditions under which the microscopic analysis was performed, in item 2.1
3. Regarding the contamination experiments, discuss its reproducibility.
4. According to the results found, what would be the main critical points for the industry to achieve the ideal cleaning procedure?
5. What would be the future steps of the proposed work to optimize the results and enable the application of this technology in most processes?
Author Response
Dear Referee,
thank you for reviewing our manuscript entitled “Investigations on Backflush Cleaning of Spent Grain-contaminated Filter Cloths Using Continuous and Pulsed Jets” for consideration in Foods.
We are happy that the manuscript met your general expectations. In the following, we want to answer your questions and remarks. Additionally, we carefully reviewed the manuscript and adapted it (marked with track change mode). Hopefully, the adaptions and the revised manuscript will suit your expectations and lead to a positive decision by you.
Sincerely yours,
Roman Werner and Dominik Geier
- I suggest that authors review the bibliographic references used, since more than half of the references were published more than ten years ago.
We have chosen the current references due to their importance for the study and the manuscript. However, we added several new publications to highlight the status quo in more detail. Here, we also extended state of the art in filter medium cleaning by other application areas (e.g., dust filter cleaning). The new added references are as follows:
Fuchs, E.; Helbig, M.; Pfister, M.; Majschak, J.-P. Erhöhung der Reinigungseffizienz bei der Cleaning-in-Place-Reinigung durch diskontinuierliche Flüssigkeitsstrahlen. Chemie Ingenieur Technik 2017, 89, 1072–1082, doi:10.1002/cite.201600105.
Stoye, H.; Köhler, H.; Mauermann, M.; Majschak, J.-P. Untersuchungen zur Steigerung der Reinigungseffizienz durch pulsierende Spritzreinigung. Chemie Ingenieur Technik 2014, 86, 707–713, doi:10.1002/cite.201300047.
Silva, L.D.; Filho, U.C.; Naves, E.A.A.; Gedraite, R. Pulsed flow in clean‐in‐place sanitization to improve hygiene and energy savings in dairy industry. Journal of Food Process Engineering 2021, 44, doi:10.1111/jfpe.13590.
Yang, J.; Kjellberg, K.; Jensen, B.B.B.; Nordkvist, M.; Gernaey, K.V.; Krühne, U. Investigation of the cleaning of egg yolk deposits from tank surfaces using continuous and pulsed flows. Food and Bioproducts Processing 2019, 113, 154–167, doi:10.1016/j.fbp.2018.10.007.
Furumoto, K.; Narita, T.; Fukasawa, T.; Ishigami, T.; Kuo, H.-P.; Huang, A.-N.; Fukui, K. Influence of pulse-jet cleaning interval on performance of compact dust collector with pleated filter. Separation and Purification Technology 2021, 279, 119688, doi:10.1016/j.seppur.2021.119688.
He, C.; Yan, C.; Tang, C.; Huang, M.; Ren, L.; Zhang, M. Nitrogen pulse jet cleaning of the pleated filter cartridge clogged with adhesive hygroscopic dusts. Process Safety and Environmental Protection 2021, 147, 430–438, doi:10.1016/j.psep.2020.08.045.
Zhang, Q.; Liu, D.; Wang, M.; Shu, Y.; Xu, H.; Chen, H. Characteristics and evaluation index of pulse-jet dust cleaning of filter cartridge. Process Safety and Environmental Protection 2022, 157, 362–374, doi:10.1016/j.psep.2021.11.028.
Li, J.; Wu, D.; Wu, Q.; Luo, M.; Li, J. Design and performance evaluation of novel colliding pulse jet for dust filter cleaning. Separation and Purification Technology 2019, 213, 101–113, doi:10.1016/j.seppur.2018.12.022.
Furthermore, we also rechecked the book references and updated them to the most actual edition.
Narziß, L.; Back, W.; Gastl, M.; Zarnkow, M. Abriss der Bierbrauerei, 8th ed.; Wiley-VCH: Weinheim, 2017, ISBN 978-3527340361.
Spellman, F.R. Handbook of Water and Wastewater Treatment Plant Operations, 2nd ed.; CRC Press: Boca Raton, 2013, ISBN 978-1420075304.
Lelieveld, H.L.M.; Mostert, M.A.; Holah, J.T. Handbook of Hygiene Control in the Food Industry; Woodhead Pub; CRC Press; Elsevier: Cambridge, England, Boca Raton, 2016, ISBN 978-1-85573-957-4.
- I would like the authors to detail the conditions under which the microscopic analysis was performed, in item 2.1.
We added an explanation of how the images in Figure 1 (mainly illustrating 2.1.) were taken. The added text is as follows: “For this purpose, the filter cloth was positioned directly under the microscope lens for frontal and side imaging. The objective was used with a magnification factor of 50 to display a sufficient degree of detail.“
- Regarding the contamination experiments, discuss its reproducibility.
We adapted the manuscript and added some further explanations concerning the reproducibility of the contamination method: “Thus, the reproducible contamination spots provided a reasonable basis for the cleaning experiments. Due to the consistently equal mash volume, the contaminated cloth areas coincided in size and height and showed similar cleaning behavior. The reproducibility was mainly achieved by the standardized contamination procedure, same time intervals, and equal particle size distribution. In particular, the applied hammer mill proved to be a significant advantage. In this grinding apparatus, the malt grains are milled into small particle sizes, which in this case contributed to a more homogeneous contamination appearance.”Besides, we also summed up the reproducibility according to the results (e.g, confidence intervals) in the conclusion section: “For industry-related food contamination based on spent grains, the method showed satisfactory reproducibility, which was also confirmed by the determined confidence intervals. The results also showed significant differences when comparing the different cleaning concepts and critical cleaning parameters.”
- According to the results found, what would be the main critical points for the industry to achieve the ideal cleaning procedure?
We extended the conclusion section: “In summary, it can be asserted that pulsed cleaning jets are preferable for mash filter cleaning. Depending on the cloth design, applying 64 pulses in a quick sequence (clocking 100 ms) and an incident flow velocity of at least 3.0 m/s can yield good cleaning results. In addition, a more apertured cloth favors cleaning with backflushes. However, this depends on the application field and the required filtration properties. Since relatively large particles are separated during mash filtration, cloths with larger mesh sizes can be used well here.”
- What would be the future steps of the proposed work to optimize the results and enable the application of this technology in most processes?
We extended the manuscript in the conclusion section: “For a final transfer of the results into industrial application, the backflushing has to be realized on a large-scale technically. Here, direct backflushing of the entire filter cloth surface with a pump system is conceivable. It is also possible to place nozzles in the filter plate behind the filter cloth. If appropriately arranged, they enable the targeted generation of backflushes with an efficient cleaning effect. The choice of the proper technique always depends on the design of the filter press, the manufacturer and the prevailing local requirements. For an efficient cleaning process, the cloth geometry applied should, in all cases, be taken into consideration in the preliminary stages. This requirement determines to a large extent which basic cleaning parameters must be used.” Besides, we are already working on an up-scale of this cleaning concept to an industrial version. Here, we are currently equipping a mash filter system in our research brewery. This system comprises plate packages of 80 cm x 80 cm, which offers commercial-close conditions.

Reviewer 2 Report
The manuscript "Investigations on Backflush Cleaning of Spent Grain-contaminated Filter Cloths Using Continuous and Pulsed Jets" is quite interesting and very specific for the brewing and potentially the distilling industries.
I have some relatively minor points to make:
Line 36: replace the word 'hold' with 'facilitate'
Line 38: replace the word "are" with "can be".
Line 38: add the words "filter surface during the" between 'the' and 'cleaning process'.
Line 44: insert the word "the" between 'from' and 'liquid wort'.
Line 46: provide a more precise statement about lauter tun filtration. Such as: "... with a slotted plate as a filter support with spent grain as the actual filter medium."
Line 47: please clarify the "and applied cereals" statement in that sentence. I struggle to understand what you are saying.
Line 57: replace the word "in" with "over"
Figure 1. Why are you showing two different magnifications? They are different cloths. Please show them at teh same magnification.
Line 127: The opening sentence is confusing. I don't understand the words "was the central use case"
Figures 4 onwards.
I know you are using different cloths, but in the legends can you please simplify the annotation for the cloths?
The symbols in the figure are very small and make it hard to "quickly" read the figures. Please increase the symbol size
Figures 7A and 7B could and should be combined for clarity. The same goes for 7C and 7D
The rest of the paper is "fair enough"
Author Response
Dear Referee,
thank you for reviewing our manuscript entitled "Investigations on Backflush Cleaning of Spent Grain-contaminated Filter Cloths Using Continuous and Pulsed Jets" for consideration in Foods.
We are happy that the manuscript met your general expectations. In the following, we want to answer your questions and remarks. Additionally, we carefully reviewed the manuscript and adapted it (marked with track change mode). Hopefully, the adaptions and the revised manuscript will suit your expectations and lead to a positive decision by you.
Sincerely yours,
Roman Werner and Dominik Geier
- Line 36: replace the word 'hold' with 'facilitate.
The manuscript was adapted according to your recommendation.
- Line 38: replace the word "are" with "can be".
The manuscript was adapted according to your recommendation.
- Line 38: add the words "filter surface during the" between 'the' and 'cleaning process'.
The manuscript was adapted according to your recommendation.
- Line 44: insert the word "the" between 'from' and 'liquid wort'.
The manuscript was adapted according to your recommendation.
- Line 46: provide a more precise statement about lauter tun filtration. Such as: "... with a slotted plate as a filter support with spent grain as the actual filter medium."
We adapted the manuscript according to your recommendation. Additionally, we explained the lauter tun's functionality and limits in more detail. The manuscript's text is now as follows: "This segregation can also be performed using lauter tuns with a slotted plate as filter support with spent grain as the actual filter medium. The plate is usually located close to the tun's bottom and facilitates dead-end filtration. Unfortunately, the lauter tun's flexibility is limited due to long filter times as the entire mash volume has to be separated using one single tun."
- Line 47: please clarify the "and applied cereals" statement in that sentence. I struggle to understand what you are saying.
Here, we meant the possible variety of applied cereals. Usually, lauter tun's are limited to malted barley, wheat or partly rye. Only using these cereals is possible as they still guarantee enough filtration performance in such a big volume without blocking the filter cake. Mash filters are more flexible as also other cereals such as corn or unmalted barley can be applied. We adapted the manuscript as follows: "Here, mash filter systems are often preferred because they are typically more flexible in batch quantities and cereal variety (e.g., unmalted barley or corn) than other systems [7]."
- Line 57: replace the word "in" with "over"
The manuscript was adapted according to your recommendation.
- Figure 1. Why are you showing two different magnifications? They are different cloths. Please show them at teh same magnification.
We adapted Figure 1 as recommended. Now, both filter cloths are shown with an amplification factor of 50. Additionally, we adapted the formatting according to the journal's requirements.
- Line 127: The opening sentence is confusing. I don't understand the words "was the central use case"
Here, we wanted to express that the cleaning study of a mash filter represented an industry-oriented application. Thus, it offered suitable conditions for carrying out our experiments in an industrial context. In previous studies, model contamination was applied frequently, which often is challenging to transport to industrial applications. We have adapted the text: "In this study, the mash filter in a brewery served as an industry-related case study that required a corresponding contamination type based on spent grains."
- Figures 4 onwards.
I know you are using different cloths, but in the legends can you please simplify the annotation for the cloths?
We had the problem that both filter types had complicated annotations from the beginning. So we decided to name them just F1 (Fitler 1) and F2 (Filter 2). The rest of the annotation also comprises the most critical characteristics like weave type and mesh size. This addition helps identify the most influencing parameters directly on backflushing concerning cloth characteristics. So, it has already been a simplification. For a better understanding, we added an explanation to section 2.1. (line 110-113) as follows: "The annotation F1 represented filter type 1 with a twill weave (TWL) and mesh sizes of 80 and 100 µm. On the other hand, filter type 2 was designated with F2 and had a satin weave (STN) and mesh sizes of 20 µm."
The symbols in the figure are very small and make it hard to "quickly" read the figures. Please increase the symbol size
We adapted Figure 4 and enlarged the symbols by reducing the scale. Initially, we wanted to display the measurement results up to a 1.0 scale. Unfortunately, this nearly made the error bars invisible, as they are tiny. We hope that the overall looking of the figure is now better and meets your expectations.
- Figures 7A and 7B could and should be combined for clarity. The same goes for 7C and 7D
We followed your recommendation and combined the figures.
